# Gestational Exposure to Bisphenol A Affects Testicular Morphology, Germ Cell Associations, and Functions of Spermatogonial Stem Cells in Male Offspring

**DOI:** 10.3390/ijms21228644

**Published:** 2020-11-17

**Authors:** Polash Chandra Karmakar, Jin Seop Ahn, Yong-Hee Kim, Sang-Eun Jung, Bang-Jin Kim, Hee-Seok Lee, Buom-Yong Ryu

**Affiliations:** 1Department of Animal Science and Technology and BET Research Institute, Chung-Ang University, Anseong 17546, Korea; polashmicro@gmail.com (P.C.K.); ahnjs@cau.ac.kr (J.S.A.); yhkcau@naver.com (Y.-H.K.); tkddms2428@naver.com (S.-E.J.); 2Department of Cancer Biology, Perelman School of Medicine, University of Pennsylvania, Philadelphia, PA 19104, USA; bakim@pennmedicine.upenn.edu; 3Department of Food Science & Technology, Chung-Ang University, Anseong 17546, Korea; hslee0515@cau.ac.kr

**Keywords:** bisphenol A, testis morphology, male germ cell, apoptosis, spermatogonial stem cell function

## Abstract

Exposure to bisphenol A (BPA) in the gestational period damages the reproductive health of offspring; detailed evidence regarding BPA-induced damage in testicular germ cells of offspring is still limited. In this study, pregnant mice (F0) were gavaged with three BPA doses (50 μg, 5 mg, and 50 mg/kg body weight (bw)/day; tolerable daily intake (TDI), no-observed-adverse-effect-level (NOAEL), and lowest-observed-adverse-effect level (LOAEL), respectively) on embryonic days 7 to 14, followed by investigation of the transgenerational effects of such exposure in male offspring. We observed that the NOAEL- and LOAEL-exposed F1 offspring had abnormalities in anogenital distance, nipple retention, and pubertal onset (days), together with differences in seminiferous epithelial stages and testis morphology. These effects were eradicated in the next F2 and F3 generations. Moreover, there was an alteration in the ratio of germ cell population and the apoptosis rate in germ cells increased in F1 offspring at the LOAEL dose. However, the total number of spermatogonia remained unchanged. Finally, a reduction in the stemness properties of spermatogonial stem cells in F1 offspring was observed upon LOAEL exposure. Therefore, we provide evidence of BPA-induced disruption of physiology and functions in male germ cells during the gestational period. This may lead to several reproductive health issues and infertility in offspring.

## 1. Introduction

Bisphenol A (2,2-bis(4-hydroxyphenyl)propane; BPA) is a ubiquitous endocrine disruptor (ED) with widespread industrial applications, especially to make plastics and epoxy resins used by consumers [1,2]. BPA is chemically stable, can persist after leaching into water and foods [3,4], and accumulates in the body [5,6]. Therefore, the presence of this compound is found in human saliva, blood, plasma, amniotic fluid, placental tissue, breast milk, urine, follicular fluid, and adipose tissue [2]. Previous studies have established that BPA has estrogenic and anti-androgenic properties both in vivo and in vitro [7,8]. As a consequence, it is able to enter into several endocrine-related pathways [9], affect the reproductive system [10], and induce toxicity on the proliferation and maintenance of several cell types, including testicular germ cells [11,12,13,14].

To maintain male fertility, spermatogonial stem cells (SSCs) undergo spermatogenesis and produce sperm through several cycles of mitosis and meiosis along with controlled apoptosis [15,16]. BPA is a xenoestrogenic compound that possibly hinders hormonal signals by reacting with estrogen receptors on testicular germ cells [17]. Therefore, BPA-related in vitro effects on the characteristics of spermatozoa [18], low sperm count [19], increased DNA damage in sperm [20], and protein alteration in sperm [21] have been studied well. Moreover, other studies demonstrating BPA-related disruption in the meiotic process [22] and reduction in the crossover of chromosomes [23] have also been conducted. Additionally, BPA-induced effects on oxidative stress [24] and DNA methylation [25] have been proven. Therefore, laws have been made that define ranges for BPA exposure: tolerable daily intake (TDI; 50 μg/kg body weight (bw)/day), no-observed-adverse-effect-level (NOAEL; 5 mg/kg bw/day), and lowest-observed-adverse-effect level (LOAEL; 50 mg/kg bw/day) [26,27].

Doyle et al. [28] reported on the effects of di(2-ethylhexyl) phthalate (one of the EDs) on the physiological characteristics and reproductive development of male mice upon exposure between gestational days 12 and 21. It has been shown that a relatively high dose of EDs can cause testicular abnormalities and anomalies in germ cell association, which can transfer transgenerationally. Although the manufacture and use of BPA are maintained within the law-provided values, BPA-induced effects are observed within these proposed levels as well. For instance, neonatal exposure to low doses of BPA can induce meiotic arrest and death of testicular germ cells [29]. Micromolar concentrations of BPA can alter functional properties and the proteome of testicular germ cells [30] and Sertoli cells [31] in mice. Moreover, studies on BPA exposure (<50 mg/kg bw/day) during the gestational period showed a reduction in the proportion of elongated spermatids in pubertal mice [32], low sperm count in rats [33], and alteration in rodent steroidogenesis [34]. A recent study has revealed that a BPA dose of <50 mg/kg bw/day can induce DNA strand breaks [35] and impair mitochondrial functions of sperm [18]. Additionally, research on BPA exposure in pregnant mice has shown destruction of functional properties and proteome profile of F1 spermatozoa in adulthood, which results in dysfunction of several cell processes and an increase in many health issues [36]. Thus, the mechanisms by which BPA affects germ cell functions and reproduction have been well studied. However, it is necessary to evaluate the transgenerational effects of BPA on testicular morphology, germ cell associations in the testis, and functional properties of SSCs if the exposure occurs during the gestational period.

Our study showed that maternal exposure to BPA from embryonic day 7 to day 14 (E7–E14) of pregnancy is able to cause transgenerational effects on testis physiology and germ cell properties of the offspring. The BPA exposure strategy was chosen between E7 and E14 because the migration of primordial germ cells starts at E8.5, changes in global DNA methylation and demethylation (in males and females) occur around E10.5–E13.5, remethylation occurs in germ cells (males) around E14.5, and sex determination occurs at E12.5 [37]. In the current study, physiological changes in F1 offspring at their prepubertal and pubertal ages were carefully examined as BPA exposure occurred during the embryonic stages. Additionally, abnormalities in the testis morphology, germ cell staging in the seminiferous epithelium (SE), germ cell proportions, and BPA-induced apoptosis in testicular cells were evaluated transgenerationally. Finally, the effects of BPA on the stemness properties of SSCs were measured by conducting a germ cell transplantation procedure.

## 2. Results

### 2.1. Effects of BPA on Embryo Survival, Anogenital Distance, and Nipple Retention in CD1 Male Pups

The experimental design is illustrated in Figure 1A. We administered different doses of BPA to pregnant female mice (E7 to E14) and examined the survival rate of the embryos at postnatal day 21 (PND 21) [28]. F1 offspring, produced from BPA-exposed dams (F0), was set to mate with age-matched nontreated females at their PND 49–59. At least 2 male offspring were selected from each dam and paired with female offspring to generate the F2 generation. F3 generation was also produced using the same scheme (Figure 1B). Anogenital distance (AGD) measurements were made in all surviving male pups (F1) at PND 1. We observed a remarkable reduction in AGD in F1 pups as a result of BPA exposure to the embryos during E7 to E14; the effect was found in case of all BPA doses (TDI, NOAEL, and LOAEL), including ethinylestradiol (EE)-exposed control (Figure 1C). Supernumerary nipples were examined at PND 10 of F1 male pups and higher frequencies were observed at the NOAEL and LOAEL doses (Table 1). We included both male and female pups (F1 offspring) and observed a significant reduction in embryo survivability in dams treated with LOAEL (Table 1). However, no significant effects were observed in F2 and F3 pups in terms of survivability, AGD, and nipple retention (Appendix A).

Additionally, we examined BPA-related effects on litter size, weaning rates of F1 male pups, and number of implantation sites in dams and observed no significant changes in these parameters (Table 1).

### 2.2. Effects of BPA on Body Weight, Pubertal Onset, and Testis Weight of F1 Male Pups

The weights of F1 male pups were measured at PND 10 to identify any BPA-induced effects. Although pups exposed to higher BPA doses showed higher body weight, the differences were not statistically significant (Figure 1D). Interestingly, we observed that F1 male offspring from LOAEL-exposed dams had significantly delayed pubertal onset than the control and other BPA-exposed groups (Figure 1E). However, no significant difference was observed in the pubertal onset of the F2 and F3 generation males. We measured the testes weight of F1 (at PNDs 30, 60, and 120), F2 (at PND 120), and F3 (at PND 120) generations during collection. The testes weight of F1 male offspring from the dams exposed to LOAEL was significantly reduced at PND 30 (Appendix A). Similarly, significant reductions in testes weight were observed in F1 male offspring in the NOAEL- and LOAEL-exposed groups at PND 60 (Appendix A). These weight reduction scenarios were recovered at PND 120, and we did not find any remarkable differences in testes weight among BPA-exposed and control F1 offspring at this time point (Appendix A). Additionally, no differences were observed in the testes’ weights of the F2 and F3 generations at PND 120 (Appendix A).

### 2.3. BPA Increases Frequency of Abnormal Seminiferous Tubules in F1 and F2 Offspring

Paraffin-embedded and hematoxylin and eosin-stained testis sections of F1, F2, and F3 offspring were examined for testicular abnormalities. Seminiferous tubules (STs) that showed a large lumen size, abnormal cell mass in the lumen area, germ cell loss with the presence of abnormal vacuoles, and no lumen were considered as STs with abnormality (Figure 2A). We observed a significantly higher percentage of abnormal STs in F1 offspring from dams exposed to all BPA doses (Figure 2B) at PND 30. This state continued until PND 60 (Figure 2C). Moreover, NOAEL- and LOAEL-exposed dams of F1 offspring showed remarkably higher frequency of ST abnormality at PND 120 (Figure 2D). Similarly, significantly higher percentages of abnormal STs were observed in both NOAEL- and LOAEL-exposed dams of F2 males at PND 120 (Figure 2E). However, we observed no significant changes in the percentage of abnormal STs in the F3 generation at PND 120 (Figure 2F). Additionally, in EE exposed group were observed significant changes in F2 and F3 generating, which could have a strong effect of abnormal STs during generation. BPA induced alterations in the count of different types of spermatogonia and germ cells.

### 2.4. BPA Affects the Area of the Seminiferous Epithelium and Alters the Stages of Seminiferous Tubule

The frequency of abnormal STs persists in F1 male mice exposed gestationally to BPA and has been found to transfer to the next generation (F2). This finding led us to examine BPA-related effects on the area of the SE and the status of spermatogenesis stages inside the ST. We measured the areas of ST and lumen of F1 (at PNDs 60 and 120), F2, and F3 at different ST stages. We did not observe any significant changes in the ST area in F1 males at PNDs 60 (Appendix A) and 120 (Appendix A), but found significantly higher lumen areas in stages VII and VIII in case of F1 from the LOAEL-exposed dams, both at PNDs 60 (Appendix A) and 120 (Appendix A). A higher lumen area actually represents a thinner area of SE among STs of the same size. Therefore, we observed a reduction in the SE area of F1 males due to gestational BPA exposure. Additionally, we examined the area of the lumen in F2 (Appendix A) and F3 (Appendix A) males at PND 120, but did not observe any changes among the treatment groups. Next, we scrutinized the stages in SE and found that there was a significant increase in the percentage of stage VII in F1 males (PND 120) from the LOAEL-exposed dams, while the percentage of stage VIII was dramatically decreased in the same group (Figure 3A). However, no significant differences in SE stages were observed in F2 (Figure 3B) and F3 (Figure 3C) males at PND 120.

### 2.5. Effects of BPA on the Number of Spermatogonia and Germ Cells

We observed that the area of the SE was reduced in F1 due to gestational BPA exposure. Therefore, we evaluated the total number of spermatogonia in comparison with the number of Sertoli cells in the same ST (Figure 4A). Interestingly, we did not observe any difference in the ratio of total spermatogonia/Sertoli in F1 males at PNDs 60 (Figure 4B) and 120 (Figure 4C). We also measured the ratio of total spermatogonia/Sertoli in F2 (Figure 4D) and F3 (Figure 4E) males at PND 120 but found the results to be consistent with those obtained in the F1 offspring.

Next, we evaluated the effect of BPA on the total number of testicular germ cell subpopulations using fluorescence-activated cell sorting (FACS). These results showed three peaks according to the DNA content (1C, 2C, and 4C cells). 1C cells typically represent spermatids, 2C cells represent somatic cells, spermatogonia, and secondary spermatocytes, and 4C cells represent cells in the G2/M phase and primary spermatocytes (Figure 5A represents germ cell population status in the control and LOAEL-exposed groups). In the F1 offspring at PND 120, we observed that the total spermatid number significantly increased in the LOAEL-exposed group, while it decreased in the NOAEL-exposed group (Figure 5B). F1 from the LOAEL group also showed a significant reduction in the number of 2C-type cells (Figure 5B). The number of 4C-type cells decreased in F1 males from both the NOAEL- and LOAEL-exposed groups (Figure 5B). Additionally, we also evaluated the composition of testicular germ cells in F2 (Figure 5C) and F3 (Figure 5D) offspring at PND 120 but observed no significant transgenerational effect of BPA among them, with respect to germ cell populations.

### 2.6. BPA Exposure Induces Germ Cell Apoptosis

Gestational BPA exposure increased the number of germ cells positive for terminal dUTP nick-end labeling (TUNEL) (Figure 6A represents the status of germ cell apoptosis in control, LOAEL-, and EE-exposed groups) in F1 males. In our experiment, we observed that the percentage of apoptotic tubules significantly increased in the LOAEL-exposed F1 offspring (at PND 120) (Figure 6B). The number of TUNEL-positive germ cells per tubule also markedly increased in the same exposure group of F1 (Figure 6E). However, germ cell loss was recovered in the next generations. We observed no significant changes in the percentage of apoptotic tubules and apoptotic cells in F2 (Figure 6C,F) and F3 (Figure 6D,G) offspring at PND 120.

### 2.7. BPA Exposure Does Not Change Litter Size of F1 and F2 Generations

We observed that gestational BPA exposure did not hamper the litter size of BPA-exposed dams (F0), but reduced the survival rate of pups in the LOAEL-exposed group (Table 1). Therefore, we measured the litter sizes generated from F1 and F2 male offspring. The number of newborn F2 pups from the F1 father decreased in the higher exposure groups, but these differences were not significant (Appendix A). Similarly, no differences were observed in the litter size from F2 males (Appendix A).

### 2.8. BPA Affects the Stemness Properties of Spermatogonial Stem Cell in F1 Offspring

We observed that gestational BPA exposure caused effects in the F1 offspring from the NOAEL- and LOAEL-exposed groups, which diminished in the next generations. Therefore, we decided to transplant germ cells from both NOAEL- and LOAEL-exposed F1 male into recipient CD1 male mice. After one month of transplantation, germ cells labeled with PKH26 cell membrane linker dye (red fluorescence) were visualized under a fluorescence microscope (Figure 7A) and donor-derived colonies were counted. We observed a significantly lower colony count in F1 male offspring exposed to LOAEL, as compared to the control group (Figure 7B). F1 males from the EE-exposed control group also showed a significant reduction in colony number, whereas no change was found in the NOAEL-exposed group (Figure 7B).

## 3. Discussion

As BPA has the ability to promote several diseases and health hazards, this xenoestrogenic compound has become a crucial topic of research over the last 50 years. Some studies have reported the vicious effects of BPA on reproduction and sexual dysfunction in males and females upon exposure in childhood or even in adulthood [38,39,40]. This naturally-persistent substance has also been reported as one of the important factors in breast cancer [41], neuroblastoma [42], thyroid dysfunction [43], heart disease [44], asthma [45], and obesity [46]. Therefore, legislation has been approved on the use of BPA. In addition, BPA-related studies are usually conducted within the FDA proposed tolerance ranges.

The toxicological significance of BPA has been reported in previous studies [47,48,49]. We investigated transgenerational effects of BPA (at TDI, NOAEL, and LOAEL doses) on testicular morphology, ST organization along with germ cell association, and functional properties of SSCs, when the exposure to the embryos occurred during the gestational period. Therefore, we evaluated alterations in neonatal physiological parameters. In our experiment, we did not observe any differences in the litter size and implantation site of F0 dams due to BPA exposure (Table 1). However, we observed a reduction in the embryo survival rates of F1 pups after LOAEL exposure. These findings indicated that gestational BPA exposure does not cause embryo loss to the mother, but can hamper the survival of offspring. Additionally, we measured weaning rates of the surviving F1 offspring and found no differences among the control and exposure groups.

As BPA exposure occurs in embryonic stages, we evaluated AGD in newborn male pups at PND 1. We found that BPA reduced the distance between the anus and genitalia in a dose-dependent manner (Figure 1C). This physiological change in male pups led us to observe the incidence of supernumerary nipple retention among males. At PND 10, we found that higher percentages of NOAEL- and LOAEL-exposed pups along with EE-exposed pups retained nipples, as compared to the control group (Table 1). However, there was no change in their body weights at PND 10 due to the effect of BPA doses (Figure 1D). Therefore, we planned to check whether there were any changes in pubertal onset among the BPA-exposed F1 offspring and started to observe penis detachment from the prepuce of pups from PND 20 onwards. As shown in Figure 1E, we observed that LOAEL exposure caused significantly delayed pubertal onset in F1 males. The above-mentioned physiological parameters were also examined for F2 and F3 generations, but no significant changes were observed. Therefore, we concluded that BPA-induced changes in the physiology of pups can be observed only when exposure occurs at the embryonic stages.

Next, we examined the effects of gestational BPA exposure on the testis weight of F1 male offspring. We observed a reduction in the weight of immature testes after exposure to LOAEL at PND 30. This scenario continued and we found significantly less weight in the case of NOAEL and LOAEL exposure at the F1 adult age (PND 60). However, these weight reductions were recovered and no differences could be observed at PND 120 (Appendix A). Moreover, male offspring of the F2 and F3 generations did not show any differences in testis weight. Therefore, we planned to examine the internal parts of the F1 testis exposed to BPA and tried to summarize the abnormalities observed in STs due to BPA exposure. As shown in Figure 2B–D, higher percentages of abnormal tubules were found at all BPA doses in F1 males, at both PND 30 and PND 60; we observed the same situation in the NOAEL- and LOAEL-exposed groups at PND 120. Therefore, we presumed that the presence of high frequencies of abnormal tubules due to BPA exposure could be a sustainable effect, although the weights of the testes were not significantly affected. Additionally, we observed higher percentages of abnormal tubules in F2 males (at PND 120) from the NOAEL- and LOAEL-exposed groups, which were recovered in the F3 generation (at PND 120).

While examining the tubular abnormalities, we observed that some of the areas of the lumen were abnormally large, thus resulting in a comparatively small area for the SE. Therefore, we examined the BPA-induced changes in the SE of F1 offspring and measured the areas of the STs and lumens. As the volume of tubules and lumens are different in different stages of SE [50], we took measurements according to tubular staging. We did not find any significant changes in the area of the tubules but observed a significantly larger lumen area at stages VII and VIII in the LOAEL-exposed group, both at PND 60 and PND 120 (Appendix A). Therefore, we also examined the lumen area at stages VII and VIII of F2 and F3 offspring (at PND 120) to determine whether this effect is retained transgenerationally. However, we observed that BPA-induced changes in the SE area were recovered in the next generations. Moreover, the BPA-related effect on the staging of SE was also evaluated at the adult age (PND 120). In F1 offspring of the LOAEL-exposed group, we observed that the percentage of tubules at stage VII increased significantly, while the percentage of stage VIII tubules decreased remarkably (Figure 3A). Although the reasons for the inconsistencies in frequencies between stages VII and VIII are unclear, we presume that a reduction in the percentage of stage VIII tubules suggests a possible delay in the process of spermiation that occurs at stage VIII [51]. Therefore, this finding can be an explanation for the BPA-induced reduction in the level of sperm production described in a previous study [22]. We also examined the staging of SE in F2 and F3 generations (at PND 120) but observed no differences (Figure 3B,C). Thus, gestational BPA exposure-related changes in SE staging appeared only in F1 offspring, and are eradicated in the next generations.

Our next aim was to determine BPA-induced changes in the proportion of germ cells as we observed testicular abnormalities and differences in stages of SE due to BPA exposure. Generally, spermatogonia are classified into type A spermatogonia, intermediate spermatogonia, and type B spermatogonia, which appear during SE stages I to VI [51]. We calculated the ratio of spermatogonia to Sertoli cells for each tubule that showed the above-mentioned stages. We did not find any difference in this ratio due to BPA exposure in F1 offspring, at either PND 60 or PND 120 (Figure 4B,C). As shown in Figure 4D,E, this situation remained similar in the F2 and F3 generations. Additionally, we measured the number of Sertoli cells from the same SE of stage I-VI because previous studies have shown that in vitro BPA exposure affects the proliferation and proteome of Sertoli cells [31]. However, no differences in the number of Sertoli cells were observed in our experiment (data not shown). Therefore, we planned to evaluate the total number of cells in spermatogenesis and spermiogenesis among the BPA-exposed and control groups using the DNA content method. In adult F1 offspring (PND 120), we observed that the number of 1C type cells (spermatids) decreased in the NOAEL-exposed groups but dramatically increased in the LOAEL-exposed groups (Figure 5B). The relatively higher number of SEs with stage VII could be the reason for the higher number of spermatids and haploid cells in the LOAEL-exposed group. Moreover, we observed that many types of the STs lacked lumen (Figure 2A). This type of tubular abnormality was higher in the BPA-exposed groups (Figure 2B–D). This could also be the reason for the higher number of 1C-type counts. However, we observed a significantly lower number of 2C-type cells (somatic cells; spermatogonia and secondary spermatocytes) along with 4C-type cells (G2/M and primary spermatocytes) due to BPA exposure (Figure 5B). As shown in Figure 5C,D, these effects were not observed in the F2 and F3 generations. Therefore, it can be stated that gestational BPA exposure can hamper the spermatogenesis process in adult F1 males, which remediate in the next generations. It is also important to determine whether there were apoptotic cells among the germ cells exposed to and affected by BPA administration. A previous study has shown that subcutaneous BPA injection in mouse pups increases the frequency of apoptotic STs at PND 22 [29]. In our experiment, embryonic BPA exposure also revealed a similar result in exposed males at adulthood. In F1 offspring at PND 120, we observed a higher percentage of apoptotic germ cells containing STs due to LOAEL exposure (Figure 6B) and a significantly higher number of apoptotic germ cells per ST in the same exposure group (Figure 6E). Therefore, gestationally administrated BPA (LOAEL) induces destruction of germ cells in male offspring and this result can also be an explanation of the possible reduction in sperm count. However, this effect was eradicated in the F2 and F3 offspring. Additionally, we checked the litter sizes from F1 and F2 offspring as we experienced a lower number of germ cell populations (Figure 5B) and loss of germ cells due to the BPA-induced effect (Figure 6B,E). Although the number of F2 pups (litter size from F1) reduced in the BPA-exposed groups (Appendix A), these differences were not significant compared to the control group. Similarly, no differences were found in the number of F3 pups from each group (Appendix A).

Finally, we aimed to scrutinize the effect of gestational BPA exposure on stemness properties of SSCs. We conducted testicular germ cell transplantation into busulfan-treated ICR mice. As most of the BPA-induced effects on ST abnormalities along with staging of SE, germ cell proportion, rate of apoptotic germ cells were observed in F1 male offspring of the NOAEL- and LOAEL-exposed groups and persisted until adulthood in F1 males, we decided to transplant F1 (PND 120) germ cells of the NOAEL- and LOAEL-exposed groups. We found a significant decrease in the number of donor germ cell-derived colony numbers in F1 males of the LOAEL-exposed group compared to the control group (Figure 7B). Therefore, it can be stated that although BPA exposure does not affect the number of spermatogonia (Figure 4B,C), it can reduce the stemness properties of SSCs among these spermatogonia.

In conclusion, this study, to the best of our knowledge, is the first comprehensive in vivo study on mice to scrutinize the effects of embryonic BPA exposure during the gestational period, in which the exposed male offspring were tested in terms of their physiology, testicular morphology, number of germ cells, and their association with the functional properties of SSCs. These parameters were checked transgenerationally. The TDI dose of BPA had no effect on male mice, whereas the NOAEL dose showed effects in certain cases, especially on the F1 offspring at their earlier ages. BPA-induced effects on almost all of the parameters were recorded in F1 offspring at the LOAEL dose. These effects could be attributed to BPA-induced damage to the testis structure, decrease in germ cell number with the stimulation of apoptosis in cells, and reduction in stemness properties of SSCs. Thus, embryonic BPA exposure, even for a short period (E7–E14), can induce destruction in an individual’s reproductive health, if it happens at the relevant doses. Therefore, considering public health issues, it is crucial to re-evaluate BPA exposure levels that are deemed to be acceptable.

## 4. Materials and Methods

### 4.1. Animals

Breeding stocks of ICR mice (CD-1; male and female) (DBL, Eunseong, Korea) were used for mating. Recipient CD-1 male mice used for the transplantation of testicular germ cells were purchased from Orient Bio Inc., Seongnam-si, Gyeonggi-do, Korea. The protocol for caring of animals as well as the experimental methods that we performed in the study were approved by the ‘Animal Care and Use Committee of Chung-Ang University’ (IACUC Number: 2016-00009, 25 January 2016) and the ‘Guide for the Care and Use of Laboratory Animals’ published by National Institutes of Health. The mice were fed a diet containing negligible amounts of soy that did not cause any soy-based effect. Epoxide mouse cages and BPA-free polysulfone drinking bottles were used to avoid basal BPA exposure. Food and water were provided ad libitum.

### 4.2. Experimental Design, BPA Exposure, and Breeding to Generate F1–F3 Offspring

Female CD-1 mice at PND 50–56 and age-matched male CD-1 were placed in the same case in a 2:1 ratio for mating. Females were checked every day for the presence of vaginal sperm plugs and plug-positive females [considered embryonic day 0 (E0)] were separated. Three reference doses of BPA (239658, Sigma-Aldrich, St. Louis, MO, USA) were used for this study: 50 µg/kg bw/day (TDI), 5 mg/kg bw/day (NOAEL), and 50 mg/kg bw/day (LOAEL). Orally-active estrogen, EE was used as positive control at a dose of 0.4 µg/kg/day [52]. BPA and EE doses were prepared by dissolving in corn oil (Sigma-Aldrich), while the control group received corn oil only [28]. Between twelve and fifteen plug-positive dams were selected for each treatment group and gavaged from gestational days E7 to E14 [28]. Body weights of the dams were calculated each day just before gavaging so that the treatment doses were administered accurately. BPA-exposed dams (F0) produced F1 offspring upon setting for mating with an outbreed female. F2 generation was produced by setting a male offspring of F1 for mating with outbred females. F3 generation was also produced in a similar manner.

### 4.3. Embryo Survival, Neonatal and Pubertal Period Analysis, Implantation Sites, and Weaning Rates

Parameters such as embryo survival, AGD, retention of supernumerary nipples in male pups, implantation sites, and weaning rates were examined according to a procedure previously described by Doyle et al., 2013 [28]; the periods of examination are illustrated in Figure 1A. Briefly, AGDs in male pups were ascertained on PND 1 by measuring the distance between the urethral and anal opening using a microscope (SZ3060, Olympus, Tokyo, Japan). Male pups at PND 10 were anesthetized, followed by the determination of the presence of nipples. Body weights of the pups were also measured on the same day. Pubertal onset of male pups was examined from PND 20. Implantation sites were checked in F0 dams after the weaning of pups. The number of spots representing fixation of embryos in the uterine wall was determined using a microscope (SZ3060, Olympus). Embryo survival was evaluated at PND 21 and compared with the implantation sites. Weaning rates were measured by considering the litter size at PND 21.

### 4.4. Collection of Testis and Determination of Testicular Abnormalities

The detailed procedure is described in Appendix A.

### 4.5. PAS-Hematoxylin Staining, Staging of Seminiferous Epithelium, and Counting of Spermatogonia

Briefly, 5-µm testis sections were stained with periodic acid-schiff (PAS) and counterstained with hematoxylin-eosin to examine the staging of SE [51] under a TE2000-U microscope (Nikon, Tokyo, Japan). The area of the ST and lumen were measured under the same microscope using NIS Elements imaging software (Nikon). The total number of spermatogonia and Sertoli cells were measured in SE of stages I–VI [51] and the spermatogonia/Sertoli cell ratio was obtained.

### 4.6. Flow Cytometric Analysis

The procedure is described in detail in Appendix A.

### 4.7. Detection of Germ Cell Apoptosis

Apoptosis analysis was carried out in testis sections (paraffin-embedded, not stained with hematoxylin-eosin) using TUNEL assay. In Situ Cell Death Detection Kit, POD (Roche, Mannheim, Germany) was used according to the manufacturer’s protocol. As apoptotic cells emitted green fluorescence, they were visualized using a fluorescence microscope (TE2000-U, Nikon). The percentage of STs with apoptotic cells was counted. The number of apoptotic cells from each of the apoptotic cell positive STs was also measured.

### 4.8. Germ Cell Transplantation for Evaluating SSCs Activity after BPA Exposure

Transplantation of germ cells into recipient testes is the only method for precise quantification of SSCs in testicular germ cell populations. Transplantation was carried out according to a previously described methodology [53,54]. Testicular germ cells were collected from control, NOAEL-, LOAEL-, and EE-exposed F1 offspring (PND 120) and transplanted into CD-1 male recipient mice. After 1 month of transplantation, donor-derived germ cell colonies were observed under a fluorescence microscope (AZ100, Nikon, Chiyoda-ku, Tokyo, Japan). The detailed procedure is described in the Appendix A.

Donor germ cells were injected into both the testes of the recipient and we used 8–10 recipient mice for each group. The number of donor-derived colonies was calculated as: Colonies/10^5^ cells transplanted = Number of colonies × 10^5^/Number of cells transplanted.

### 4.9. Statistics

One-way analysis of variance was used to analyze the data using SPSS (version 23.0, IBM, Armonk, NY, USA) and Prism (version 5.03; GraphPad, La Jolla, CA, USA) software. Significant differences among the mean values were determined using Tukey’s Honestly Significant Difference test, where *p* < 0.05 was considered significant.

## Figures and Tables

**Figure 1 ijms-21-08644-f001:**
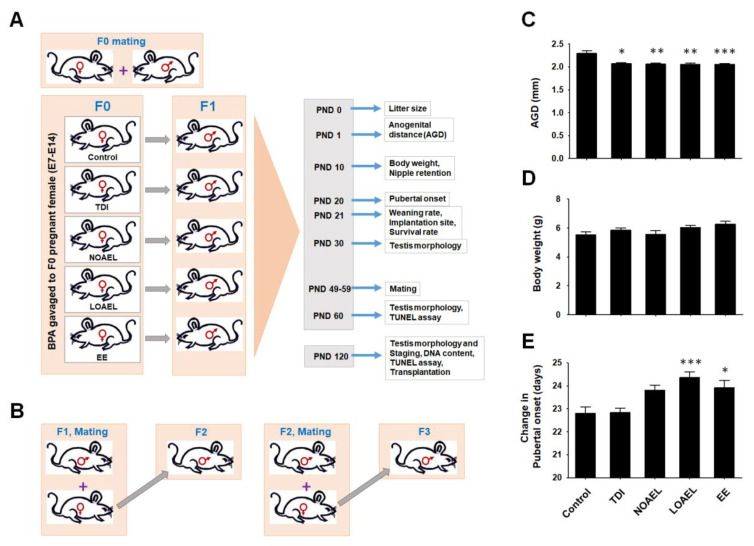
Experimental design (**A**,**B**), anogenital distance (**C**), body weight (**D**), and pubertal onset changes (**E**) in F1 male offspring. (**A**) After mating, pregnant female mice were gavaged with different bisphenol A (BPA) doses [tolerable daily intake (TDI) (50 µg/kg bw/day), no-observed-adverse-effect-level (NOAEL) (5 mg/kg bw/day), and lowest-observed-adverse-effect-level (LOAEL) (50 mg/kg bw/day); see detailed dose description in text], control (corn oil), and positive control (ethynylestradiol; EE). (**B**) Mating scheme to produce descendants (F1 to F3 generation). Age-matched nontreated females were used (PND 49–59). No inbreeding was performed. (**C**) Anogenital distance (AGD) of F1 male pups at PND 1, measured in millimeters. (**D**) Body weight of F1 male pups at PND 10, measured in grams. (**E**) Changes in days of pubertal onset for F1 offspring. Data (**C**–**E**) were generated from 23 male offspring per group. Data analyzed using one-way analysis of variance (ANOVA) where asterisk (*) indicate significant differences compared with control (* *p* < 0.05, ** *p* < 0.01, and *** *p* < 0.001).

**Figure 2 ijms-21-08644-f002:**
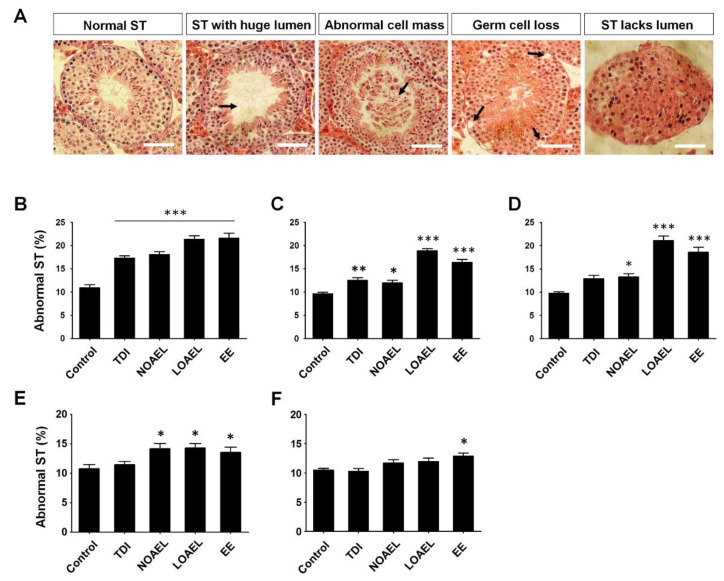
Testicular cross section and the presence of abnormal seminiferous tubules (STs) due to bisphenol A (BPA) exposure. (**A**) Hematoxylin and eosin-stained testicular cross-sections presenting normal STs along with abnormal STs, characterized by a huge lumen (>40 µm in diameter), abnormal cell mass, loss of germ cells, and lack of lumen (pictures represent STs sections at PND 120). Arrows in the images indicate the positions of abnormality. Scale bar = 50 µm. Percentages of abnormal tubules are presented as bar graphs; (**B**) F1 offspring at PND 30 (n = 15 mice/group), (**C**) F1 offspring at PND 60 (n = 15 mice/group), (**D**) F1 offspring at PND 120 (n = 15 mice/group), (**E**) F2 offspring at PND 120 (n = 20 mice/group), and (**F**) F3 offspring at PND 120 (n = 15 mice/group). Data analyzed using one-way analysis of variance (ANOVA) where asterisk (*) indicate significant differences compared with control (* *p* < 0.05, ** *p* < 0.01, and *** *p* < 0.001).

**Figure 3 ijms-21-08644-f003:**
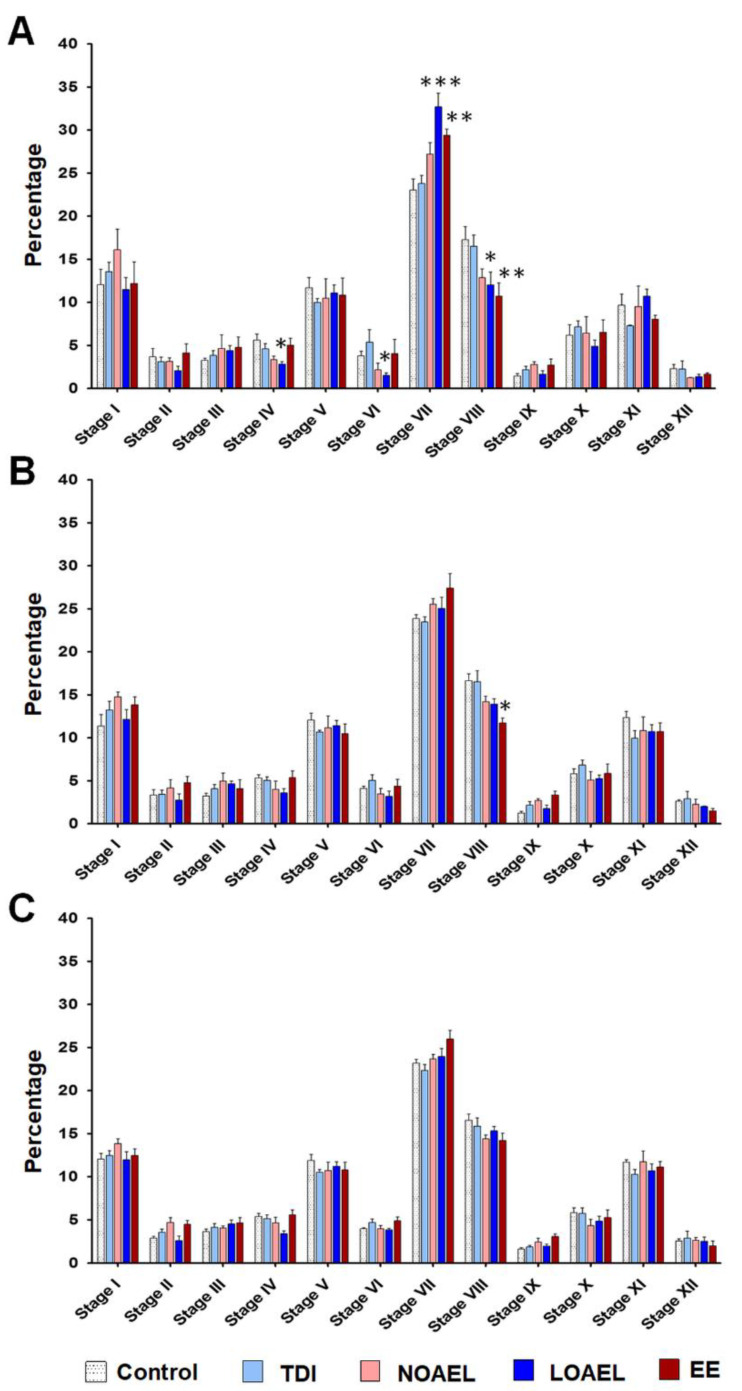
Staging of the seminiferous epithelium (SE). Testicular sections were stained with periodic acid-schiff (PAS)-hematoxylin, followed by staging of the SEs to evaluate differences in SE staging upon BPA exposure. The bar graphs represent the staging patterns of (**A**) F1 offspring at postnatal day (PND) 120 (n = 20 mice/group), (**B**) F2 offspring at PND 120 (n = 15 mice/group), and (**C**) F3 offspring at PND 120 (n = 15 mice/group). For each stage, one-way analysis of variance (ANOVA) was used to identify significant differences between exposure and control groups (* *p* < 0.05, ** *p* < 0.01, and *** *p* < 0.001).

**Figure 4 ijms-21-08644-f004:**
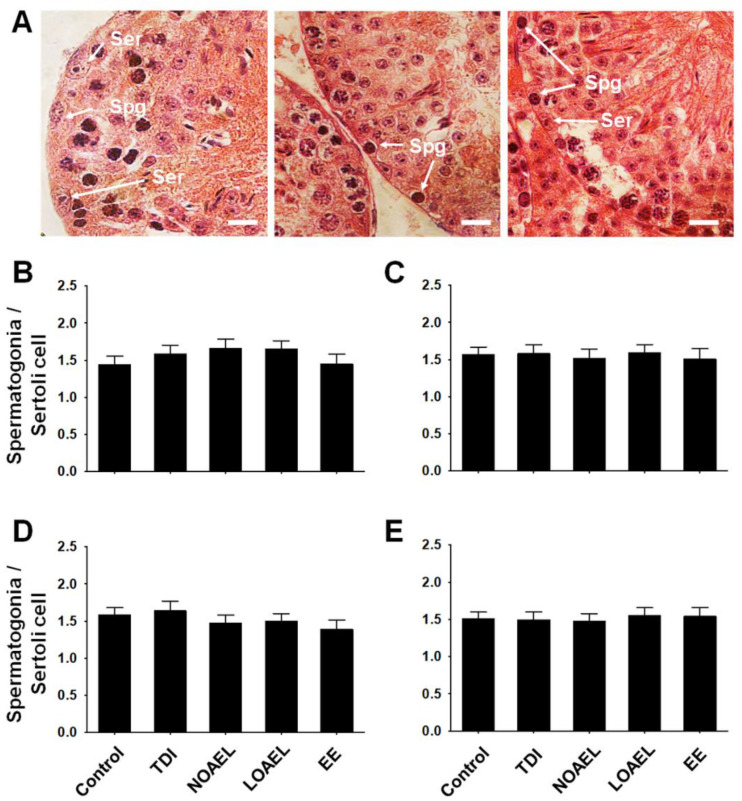
Ratio of spermatogonia and Sertoli cells. The ratio of total number of spermatogonia and Sertoli cells in the same SE was measured. (**A**) Testicular cross-sections show spermatogonia (Spg) and Sertoli cells (Ser), which usually appeared at stages I-VI. Scale bar = 10 µm. Bar graphs represent the ratios of spermatogonia and Sertoli cells of (**B**) F1 offspring at PND 60 (n = 12 mice/group), (**C**) F1 offspring at PND 120 (n = 15 mice/group), (**D**) F2 offspring at PND 120 (n = 15 mice/group), and (**E**) F3 offspring at PND 120 (n = 15 mice/group), respectively.

**Figure 5 ijms-21-08644-f005:**
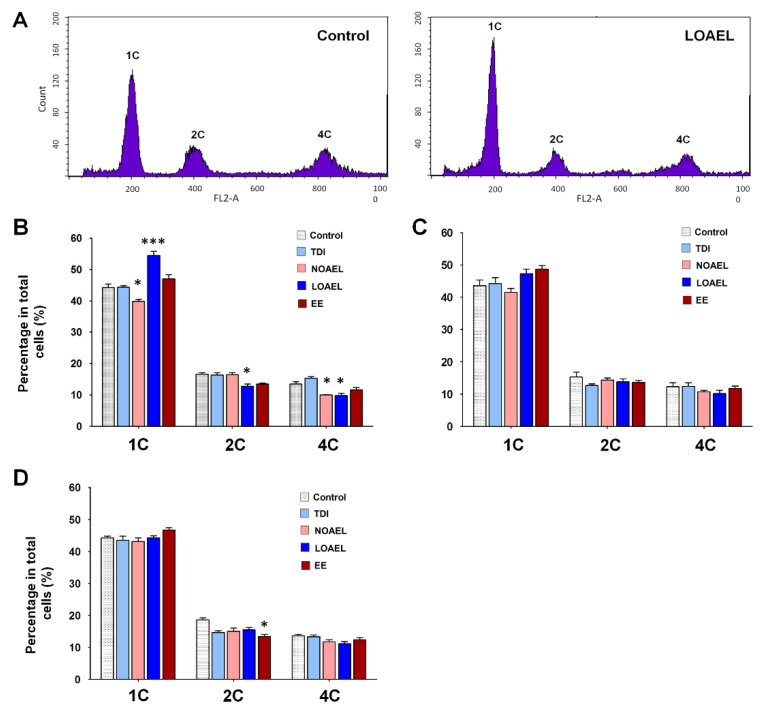
Changes in the proportion of testicular cell populations upon bisphenol A (BPA) exposure. (**A**) Representative images of control and lowest-observed-adverse-effect-level (LOEAL) doses of BPA, derived using fluorescence-activated cell sorting (FACS). 1C, 2C, and 4C indicate the proportion of haploid (1C; total spermatid), diploid (2C; spermatogonia and secondary spermatocytes), and tetraploid (4C; primary spermatocytes and G2/M cells) germ cells in the testicle, respectively. (**B**), Bar graphs represent the percentage of haploid (1C), diploid (2C), and tetraploid (4C) germ cells among the total cells of F1 offspring (PND 120). (**C**,**D**) Percentage of germ cells (1C, 2C, and 4C) of F2 and F3 offspring are represented, respectively. Data were analyzed using one-way analysis of variance (ANOVA) where significant differences (*) have been calculated upon comparison of exposure groups with control group (n = 10 mice/group; * *p* < 0.05 and *** *p* < 0.001).

**Figure 6 ijms-21-08644-f006:**
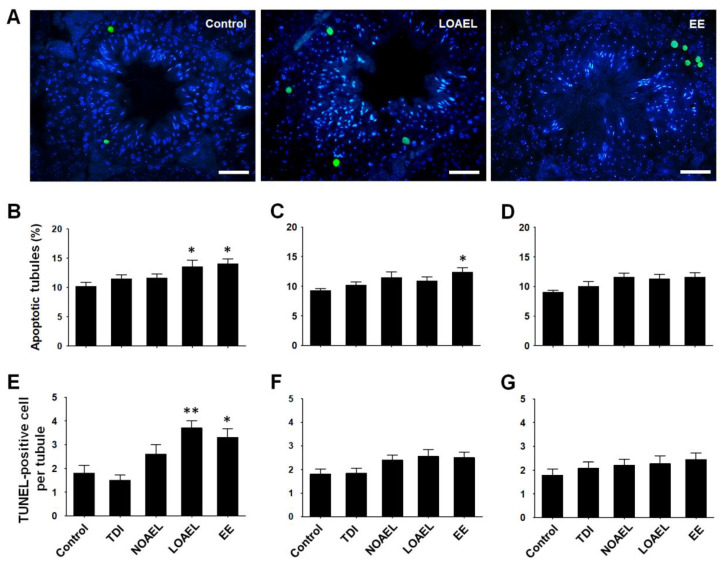
Bisphenol A-induced apoptosis in testicular germ cells. Terminal dUTP nick-end labeling (TUNEL) assay was performed to detect apoptotic germ cells in seminiferous epithelium (SE). (**A**) Images represent the position and number of TUNEL-positive germ cells (green) among other cells (blue; DAPI-stained) of SE from control, lowest-observed-adverse-effect level (LOAEL)-exposed, and ethinylestradiol (EE) groups. Scale bars = 50 µm. Bar graphs represent the percentages of apoptotic tubules in (**B**) F1, (**C**) F2, and (**D**) F3 offspring at their PND 120. Similarly, graphical presentation of TUNEL-positive germ cells per tubule in (**E**) F1, (**F**) F2, and (**G**) F3 offspring are also shown. Data were generated from at least 15 mice/group and analyzed using one-way analysis of variance (ANOVA) where * *p* < 0.05 and ** *p* < 0.01, compared to control.

**Figure 7 ijms-21-08644-f007:**
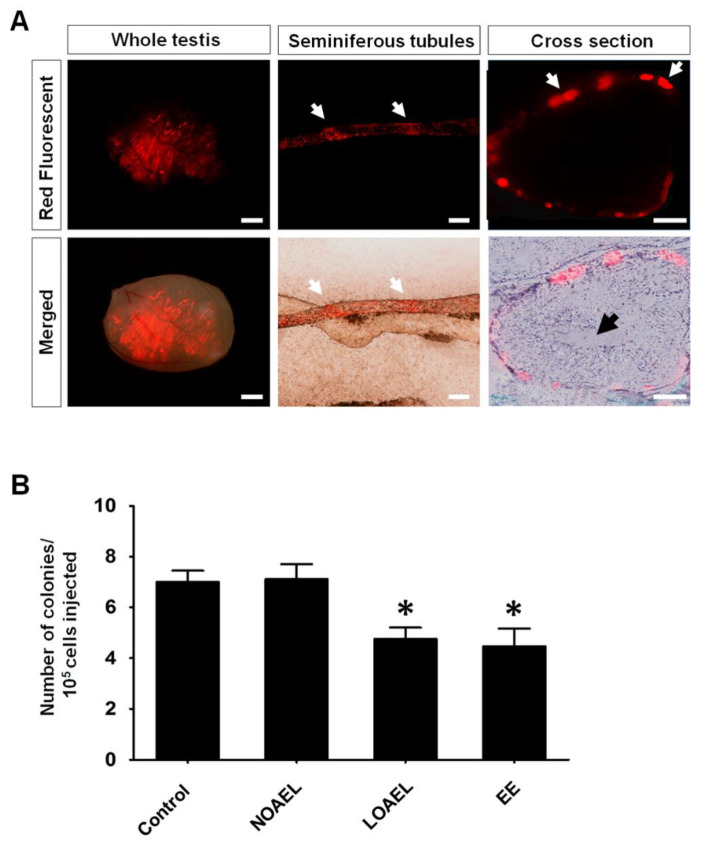
Transplantation of germ cells into recipient mice and analysis of recipient testes. Germ cell transplantation was carried out to evaluate the (bisphenol A) BPA-induced effect on the stemness properties of spermatogonial stem cells (SSCs). Donor germ cells collected from F1 postnatal day (PND) 120 offspring were stained with PKH26 (red fluorescence). (**A**) Images in the left panel show fluorescence expression of seminiferous tubules (STs) and merged image of STs inside the recipient testis (scale bar = 2 mm). The middle panel represents fluorescence and merged view of the colonization of donor germ cells inside the recipient ST. White arrows represent the places of colonization, scale bar = 200 µm. The right panel represents a cross-section of the fluorescence and merged views of the colonization of donor spermatogonia (red) on the basement membrane (white arrows) of the recipient ST (black arrow indicates sperm tails generated from donor SSCs; scale bar = 40 µm). (**B**) Bar graph represents the number of donor cell-derived colonies per 10^5^ transplanted germ cells. Total numbers of mice/testes analyzed were 8/14, 8/15, 10/18, and 8/14 for control, no-observed-adverse-effect-level (NOAEL)-, lowest-observed-adverse-effect-level (LOAEL)-, and ethinylestradiol (EE)-exposed groups, respectively. Asterisk (*) indicates significant differences (* *p* < 0.05) compared to the control.

**Table 1 ijms-21-08644-t001:** Litter size, implantation sites, and weaning rates of F0 dam, and nipple retention and survival rates of F1 offspring. *

Treatment Group	Pregnant Mice (n)	Male Pups (n)	Litter Size	Nipple Retention (%) [PND 10] **	Implantation Sites [PND 21]	Survival Rates (%) [PND 21]	Weaning Rates (%) [PND 21]
Control	12	75	12.84 ± 0.44	4.90 ± 0.88 ^b^	13.12 ± 0.55	97.15 ± 1.31 ^a^	94.21 ± 2.36
TDI	12	73	13.07 ± 0.66	7.20 ± 1.50 ^b^	13.98 ± 1.83	94.16 ± 2.71 ^ab^	89.74 ± 1.81
NOAEL	13	77	12.88 ± 1.10	14.25 ± 1.60 ^a^	14.35 ± 1.33	92.75 ± 3.01 ^ab^	93.07 ± 1.93
LOAEL	14	82	12.92 ± 0.30	13.32 ± 1.88 ^a^	14.88 ± 0.38	89.67 ± 3.16 ^b^	93.43 ± 2.91
EE control	11	74	12.60 ± 0.33	12.94 ± 1.66 ^a^	13.35 ± 0.33	94.52 ± 1.78 ^ab^	92.45 ± 2.70

* Data were analyzed using one-way analysis of variance (ANOVA), where the differences letters (a, b, and ab) separated the statistical groups. ** Data generated from male pups only.

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
