# Peer review of "Gestational Exposure to Bisphenol A Affects Testicular Morphology, Germ Cell Associations, and Functions of Spermatogonial Stem Cells in Male Offspring"

_ijms, 2020, doi:10.3390/ijms21228644_

Round 1

Reviewer 1 Report

This document is generally well written. The authors have made an adequate introduction in which they reflect updated literature. The materials and methodologies used are appropriately detailed in the experimental design. The statistical treatment of the data is well suited for this type of work. The results, expressed in the different tables, allow us to see the modifications that are presented in the different measured parameters and the values obtained in the different times of recovery. Nevertheless, it is necessary to make corrections to the Language.

Corrections in English should be made on lines:

Line 93: The verb “were” does not seem to agree with the subject. Please consider changing the verb form.

Line 94: The verb “outbreed” may be in the wrong form after the preposition with. Please consider changing it to the gerund form.

Line 104: The past participle verb “shown” has been used without an auxiliary verb. Please consider adding one or using the past simple instead.

Table 1: The words “treatmen” as well as “implantatio n” are not well written.

Line 263: The word “postnantal” is not correct. Please consider changing.

Line 481: Abbreviations; the word “epothelium” is not correct. Please consider changing

Author Response

Response:

We would like to thank the reviewer for the positive comments about our study.

Corrections in English should be made on lines:
Line 93: The verb “were” does not seem to agree with the subject. Please consider changing the verb form.

Response: Thank you for the comment. The verb form has been changed. Please refer to revised line 93.

Line 94: The verb “outbreed” may be in the wrong form after the preposition with. Please consider changing it to the gerund form.

Response: We have revised the line. Please refer to revised line 94. The breeding procedure is explained in the materials and method section.

Line 104: The past participle verb “shown” has been used without an auxiliary verb. Please consider adding one or using the past simple instead.

Response: We have changed “supplementary Table S1” and added information of F2 and F3 male offspring to make it more understandable.

Table 1: The words “treatmen” as well as “implantatio n” are not well written.

Response: The words has been written properly. Please refer to revised table 1.

Line 263: The word “postnantal” is not correct. Please consider changing.

Response: The word has been corrected. Please refer to revised line 297.

Line 481: Abbreviations; the word “epothelium” is not correct. Please consider changing

Response: The word has been corrected. Please refer to revised line 520.

Reviewer 2 Report

The authors investigate the role BPA on spermatogenesis and spermatogonial stem cells (SSCs). Here they analysed the effect of different concentrations of BPA on male animals of F2 and F3.

They measured different parameters, however, there is no clear effect to the in F2 or F3 animals. From the data the authors provided here it is difficult to draw a conclusion.

Minor points:

BPA and reduced AGD in F1, this has been shown before (Sun et al., 2018)

There is no significant difference was observed in the pubertal onset of the F2 and F3 generation males. In addition, there are no significant effects observed in F2 and F3 pups in terms of survival and nipple retention.

Fig 2A:  the authors did not indicate treatment or controls! It is unclear what the authors want to show!

Fig 2: The effect on F2 and F3 is not consistent? What does it mean? There is no effect after ethinylestradiol (EE)treatment in F2 but in F3; what does this mean?

Fig. 5 there are no data for F2 and F3!

Fig. 7: TDI is missing. There are no data for F2 und F3; does this have an impact on F2 and F3

The figure legends need improvement; they lack substantial information.

Author Response

Response:

We would like to thank the reviewer for the positive comments about our study.  These positive comments by an outside observer demonstrate to us the importance of this work.  We also feel that this reviewer’s constructive suggestions will improve the manuscript.

The authors investigate the role BPA on spermatogenesis and spermatogonial stem cells (SSCs). Here they analysed the effect of different concentrations of BPA on male animals of F2 and F3.

They measured different parameters, however, there is no clear effect to the in F2 or F3 animals. From the data the authors provided here it is difficult to draw a conclusion.

Response:

In the study we have shown the effects of embryonic BPA exposure during the gestational period, in which the exposed male offspring were tested in terms of their physiology, testicular morphology, number of germ cells, and their association with the functional properties of SSCs. These parameters have been checked transgenerationally (F1 to F3 generations). F1 offspring has received the BPA doses (TDI, NOAEL and LOAEL) during their embryonic period (E7-E14) and most of the BPA-induced effects have been observed on F1 male offspring, especially at NOAEL and LOAEL doses. The TDI dose of BPA has no adverse effect on F1 males. We have also produced F2 generations to observe whether the BPA-induced effects could persist transgenerationally. However, we have not observed any effect on F2 generation. It means that BPA-induced effects become disappear in the F2 generation. Furthermore, we have produced F3 generation to examine if there remains any possibility of recurrence of BPA-induced effects among F3 males. But we have observed no effects on F3 males. So, our experiment concluded with the findings that embryonic BPA exposure with the relevant doses, even for a short period (E7-E14), can induce destruction on an individual’s reproductive health. However, these effects recovered transgenerationally and do not transfer to the next generations.

Minor points:
BPA and reduced AGD in F1, this has been shown before (Sun et al., 2018)
There is no significant difference was observed in the pubertal onset of the F2 and F3 generation males. In addition, there are no significant effects observed in F2 and F3 pups in terms of survival and nipple retention.

Response:  We have added the survival and nipple retention of F2 and F3 male offspring to make it more understandable. Please refer to revised supplementary table S1.

Fig 2A:  the authors did not indicate treatment or controls! It is unclear what the authors want to show!

Response: We have revised the STs pictures of Fig 2A. These pictures are showing the testicular cross-sections that include normal STs and abnormal STs (characterized by a huge lumen, abnormal cell mass, loss of germ cells, and lack of lumen). These pictures are representing the STs sections of control and BPA exposed groups at PND 120 [they are not indicating the control or treatment groups]. We have added the reference for the histological section analysis in supplementary methods. Please refer to revised line 32 in supplementary data.

Fig 2: The effect on F2 and F3 is not consistent? What does it mean? There is no effect after ethinylestradiol (EE)treatment in F2 but in F3; what does this mean?

Response: Thank you for the comment. We have revised the data and found that ethinylestradiol has significant effect in F2 generation. The previous graph has been replaced by the new one.

Fig. 5 there are no data for F2 and F3!

Response: We have given the data for F2 and F3 (Fig 5C and D respectively). We have revised the Fig 5B, B’ and B” to make it more understandable. Please refer to revised Fig 5.

Fig. 7: TDI is missing. There are no data for F2 und F3; does this have an impact on F2 and F3

Response:

We decided to transplant F1 (PND 120) germ cells (NOAEL- and LOAEL-exposed groups) to the recipient mice because most of the BPA-induced effects such as ST abnormalities, staging of SE, germ cell proportion, rate of apoptotic germ cells were observed in F1 offspring in case of  BPA exposure at NOAEL and LOAEL level. TDI showed significant differences only in some of the parameters (AGD, Abnormal STs) at the early ages of mice which eventually demolished at their adult ages. So, TDI were not use in the transplantation experiment. We would like to add that, we observed significant difference at only on LOAEL- and EE-exposed groups of F1generation (no effect found on NOAEL group). Please refer to revised line 285. Moreover, gestational BPA exposure certainly affected F1, but the effects have been recovered at F2 and F3 generations. So, we did not use BPA exposed group from F2 and F3 for this experiment.  

The figure legends need improvement; they lack substantial information.

Response: We tried our level best to improve figure legends. 

Round 2

Reviewer 2 Report

From the literature it is known that BPA treatment during pregnancy reduced fertility of offsprings Wang et al., 2014.

The authors claim that there is an effect on SSC. However this is a very mild effect and there have to be more data on the mechanism.

Wei Wang, et al., (2014)

Toxicol Appl Pharmacol.

In utero bisphenol A exposure disrupts germ cell nest breakdown and reduces fertility with age in the mouse